# Individuals with Sickle Cell Disease Using SBAR as a Communication Tool: A Pilot Study

**DOI:** 10.3390/ijerph192113817

**Published:** 2022-10-24

**Authors:** Deborah M. Jean-Baptiste, Maureen Wassef, Susan Sullivan Bolyai, Coretta Jenerette

**Affiliations:** 1Tan Chingfen Graduate School of Nursing, Umass Chan Medical School, Worcester, MA 01655, USA; 2College of Nursing, University of South Carolina, 1601 Greene Street, Columbia, SC 29208, USA

**Keywords:** SBAR, sickle cell crisis, communication, inter-rater reliability of qualitative data

## Abstract

Background: Sickle cell disease (SCD) is a hemoglobinopathy that causes debilitating pain. Patients often report dissatisfaction during care seeking for pain or a sickle cell crisis (SCC). The Theory of Self-Care Management for SCD conceptualizes assertive communication as a self-care management resource that improves healthcare outcomes. Objectives: This pilot study aimed to determine whether adults with SCD could learn to use the Situation, Background, Assessment, Recommendation (SBAR) communication method using a web-based trainer, and it aimed to determine their perceptions of the training. Methods: The participants included *n* = 18 adults with SCD. Inter-rater reliability (IRR) among three reviewers was used to evaluate the participants’ ability to respond as expected to prompts using SBAR communication within the web-based platform. Content analysis was used to describe the participants’ perspectives of the acceptability of using the SBAR patient–HCP communication simulation. Results: The SBAR IRR ranged from 64 to 94%, with 72% to 94% of the responses being evaluated as the using of the SBAR component as expected. The predominant themes identified were (1) Patient–Provider Communication and Interaction; (2) Patients want to be Heard and Believed; (3) Accuracy of the ED Experience and Incorporating the Uniqueness of each Patient; and (4) the Overall Usefulness of the Video Trainer emerging. Conclusions: This pilot study supported the usefulness and acceptability of a web-based intervention in training adults with SCD to use SBAR to enhance patient–HCP communication. Enhancing communication may mitigate the barriers that individuals with SCD encounter during care seeking and improve the outcomes. Additional studies with larger samples need to be conducted.

## 1. Introduction

Sickle cell disease (SCD) is a congenital hemoglobinopathy disorder characterized by periodic episodes of acute pain [1]. It is caused by mutations in the gene encoding the hemoglobin subunit β. This mutation causes blood cells to form a sickle shape rather than the usual round cell. This sickling makes vascular circulation difficult, which results in pain [1]. Although a rare disease, SCD is one of the most common genetic diseases in the United States. SCD affects approximately 100,000 Americans [2]. African Americans have the highest prevalence of SCD. About one in 500 African Americans is affected, with about one in 12 African Americans carrying the autosomal recessive mutation [2,3].

SCD affects almost every body system. Acute pain is the hallmark of this disease, secondary to vaso-occlusion, which causes decreased oxygen to body systems. Vaso-occlusion can lead to blood clots, infections, anemia, and organ failure. Chronic manifestations include long-term organ system damage, resulting in organ dysfunction over the patient’s lifespan [1]. These chronic manifestations are the leading cause of morbidity and mortality in this disease.

SCD typically presents as chronic pain exacerbated by acute episodes known as sickle cell crisis (SCC), requiring pain management [3,4]. However, patients who present in the emergency department (ED) during a pain crisis frequently report delays in care [5,6]. This delay in care often stems from negative past treatment experiences and HCP bias. These patients may be perceived as “drug-seeking”, and their reports of pain are not believed. These factors can delay care and increase the likelihood of hospitalization [7,8].

During an evolving SCC, patients must communicate their symptoms and needs to their HCP. Communication may be difficult due to pain, but it may also be due to bias and health-related stigma during past care-seeking experiences.

### 1.1. Bias and Stigma

The literature suggests that racial biases play a significant role in this population’s care [9]. A systematic review by Bulgin et al. [10] illustrated that patients with SCD perceived stigma which interfered with their psychological and physiological well-being, had negative social impacts, and impaired the interactions between the patients and the HCP. The review also revealed that the stigma was influenced by disease status, opioid use, racism, and sociodemographics. These instances promoted negative consequences, such as decreased patient well-being and poor disease care and management [10].

Patients with SCD overwhelmingly report biases and stigma when seeking care for their illness. They report the stress, sadness, and anger associated with their interactions with HCPs as secondary to racial bias [9]. The health-related stigma of individuals with SCD can be related to discrediting pain reports, labeling and stereotyping, blaming patients for not improving their health, discrimination, racism, inadequate pain assessment, and delays in care [11]. The behavior of these patients may also be perceived as being pain medicine or drug-seeking by HCPs with negatively biased attitudes surrounding pain and pain management, leading to improper assessments and management of these patients. As a result, patients with SCD report discrimination by HCPs, such as poor treatment, and feel that their concerns are not prioritized [11,12]. Negative interaction with HCPs can also lead to adaptive care seeking, such as silence or aggressive behaviors [11]. Individuals with SCD report delaying care due to past experiences with HCPs [7].

From the perspective of the HCP, a survey of 111 HCPs [13] reported that perceived barriers to care included patient behavior (67.57%), the opioid epidemic (67.57%), overcrowding (64.86%), and concern about addiction (49.55%). At the same time, a focus group of 13 HCPs identified the barriers of high patient volumes, lack of SCD care protocols, poor communication among providers, and stigma [13]. Although not the focus of this paper, the role of race in some of these perceived barriers must be considered, given that most people in the US with SCD are Black [2]. Topics such as racial bias in caring for patients with SCD [14], roots of distrust in healthcare due to racism [15], and racialized disparities in pain management [16] are important considerations.

Despite the racial challenges, nurses and other HCPs play a significant role in advocating for patients with SCD [14]. However, for HCPs to be the best advocates for individuals with SCD, the patients with SCD need to communicate their needs to decrease the bias. Communication with HCPs is a vital part of care seeking. Importantly, it may lead to more timely and appropriate treatment while also developing a helping, trusting relationship. Therefore, by providing individuals with SCD with more efficient and effective communication skills, they may be better able to self-advocate and to feel less victimized. The HCP is also better able to understand, communicate with, and care for these patients while seeing the patient with SCD as more credible. 

### 1.2. Communication

Many patients with SCD report dissatisfaction with the quality of care they receive [6,8,17,18]. Communication with their HCP has been identified as an essential element of quality care among patients who have SCD [17]. Collins and colleagues note that using pain scales without a relational context can further exacerbate problems in securing appropriate pain relief for individuals with SCD [19]. The literature suggests that poor communication between this patient population and HCPs may be due to various factors. These may include the provider’s lack of knowledge about the disease progression and the lack of objective assessment findings, including the subjective aspects of pain perception. An additional factor may revolve around possible stereotypical perceptions within the healthcare system. One common bias about this patient population includes pain medicine-seeking behavior (referred to as drug-seeking behavior), which also prevents effective communication [17]. Patients with SCD must articulate their healthcare needs to the provider, especially in acute scenarios. Communication may improve the delay in care seeking. It also potentially impacts healthcare outcomes. A gap in the current literature exists on the communication between patients and their HCPs, especially in acute or emergent situations.

### 1.3. Situation, Background, Assessment, Recommendation (SBAR)

SBAR (situation, background, assessment, recommendation) is widely used among HCPs to communicate patient status [20,21,22]. This tool offers a standardized technique that promotes safe and effective communication that occurs in a timely fashion. SBAR is an evidence-based tool that allows clear and effective communication regarding patient conditions [21].

SBAR is a potentially helpful communication tool for patients, especially those with frequent interactions with HCPs, because it is collaborative, provides structure and technique, and offers valuable information to inform timely, appropriate care [23]. In a systematic review of SBAR on patient safety, Müller and colleagues [22] found moderate evidence for improved patient safety using this widely used communication tool.

The SBAR communication tool has recently been investigated to streamline communication between patients and their providers. The study PI was among the first to use SBAR for patient communication [23]. Few studies have been found to use SBAR as a tool to facilitate communication between patients and HCP. Clochesy, Dolansky, Hickman Jr., and Gittner [20] used focus groups to identify patients’ communication strategies with HCPs. These strategies, referred to as SBAR3, could enhance communication between patients and HCPs. Denham [21] has anecdotally discussed using SBAR to facilitate patient and HCP communication.

Safety for patients with SCD may be improved with SBAR. These outcomes may be related to their psychological health and the care-seeking experience. There are several approaches to enhancing provider communication. In an integrated review of communication between physicians and nurses in the intensive care unit, Wang and colleagues [24] discussed the use of communication tools/checklists, team training, multidisciplinary structured work shift evaluation, and electronic SBAR documentation templates to improve communication. Another communication technique that may be useful to patients, especially those who experience health-related stigmatization, is PAAIL. PAAIL is a conversational technique (Preview, Advocacy1, Advocacy2, Inquiry, and Listen) that can help minimize stress and anxiety and reduce incivility [25].

SBAR was selected for the current pilot study because it is simple and widely used among HCPs. More importantly, it is ideal for the patient to provide health-related information to guide patient-centered care during a collaborative patient–provider interaction. Additionally, SBAR may help to provide added context that includes and goes beyond the pain score [19].

### 1.4. Theoretical Framework

The framework (Figure 1) that guides this study is the revised Theory of Self-Care Management for Sickle Cell Disease (SCMSCD), published in 2014 [7]. This theoretical framework will guide understanding of the role that self-care and communication play in the management of SCD. The SCMSCD describes how vulnerability factors and self-care management resources can affect outcomes. The vulnerability factors include complications, lack of sickle cell crisis cue recognition/response, crises per year, and overprotection. The self-care management resources include assertive communication skills, coping behaviors, self-care ability, self-care actions, self-efficacy, and social support. The health outcomes include health-related quality of life, depressive symptoms, self-esteem, pain management experience, and health-related stigma. According to the theory, vulnerability factors have a negative effect on self-care management resources and health outcomes. Conversely, health outcomes are positively impacted by self-care management resources. SCMSCD supports the hypothesis that self-care management influences the relationship between vulnerability and health outcomes [7,26].

This theory has been used in the literature to support self-care and its influence on health outcomes in this population. For example, Mathie and colleagues [27] used SCMSCD to explore the experiences of Black adults with SCD during the COVID-19 pandemic. In another study, Fowora used SCDSCD to guide a dissertation examining adherence to self-care management in caregivers [28]. A study by Curtis et al. [29] discussed the usefulness of a medication administration app for children with SCD. It used this theory to form participant questions regarding their perceptions of SCD and its management.

In the SCMSCD, assertive communication skills are identified as a self-care management resource that can positively mediate health outcomes in individuals with SCD. Based on this theoretical premise, providing patients with tools to promote communication with their HCPs may positively impact their health outcomes. This pilot study focused on communicating using SBAR.

### 1.5. Purpose of the Current Study

Developing an organized, theory-based method of communication for patients with SCD to use with HCPs is vital to ensure optimal healthcare outcomes. This pilot study aimed to determine the usefulness of the SBAR-cued web-based communication skills training and to address the study participants’ perceptions of the training.

The specific aims were the following:

Aim 1: Evaluate the usefulness and accuracy of the participants in answering prompts of the SBAR-cued communication responses;Aim 2: Describe the individuals’ perspectives on the acceptability of using the SBAR patient–HCP communication simulation to better prepare for ED visits during an SCC.

This pilot study was a portion of a larger parent study.

### 1.6. Description of Parent Study

The parent study, Developing a Virtual Training Technology to Enhance Patient–Provider Communication, trained the participants to use SBAR to communicate with HCPs. It used a quasi-experimental design that employed a web-based platform to teach patients about SBAR; the patients then practiced using SBAR in a simulated healthcare encounter in the emergency department and inpatient settings. The aims of the parent study included the following:Train patients with the skills to improve patient–provider communication to meet patient needs;Pilot and evaluate the web-based learning and coaching program with at least 50 participants.

The Theory of Self-Care Management for Sickle Cell Disease (SCMSCD) [7] guided the parent study. After ethics approval, the convenience sample was recruited from a southeastern outpatient sickle cell program with a roster of approximately 400 patients. A HIPAA waiver allowed the research team to review the clinic roster to plan recruitment. The outpatient clinic staff screened potential subjects before the PI or a trained research assistant approached them in a private area of the clinic to explain the study and recruit them. The inclusion criteria included (a) a diagnosis of sickle cell disease, (b) age of 18 years or older, and (c) the ability to read, write, and understand English. All the patients who met the inclusion criteria were included until the study was closed.

Study recruitment was from August 2017 to December 2017 and occurred weekly during outpatient clinic appointments. On average, 15 patients would be scheduled for each clinic day. Approximately, 10 would be eligible for the study. This number decreased over time as some patients would have already completed the study. About 80% of the patients approached agreed to be in the study. On average, six patients would be approached as the research assistant or PI sometimes missed an eligible patient because they were recruiting or doing the intervention with another participant. The reasons for declining that were provided included not having time due to multiple appointments scheduled on the same day, transportation challenges, or pain. During recruitment, the study was described, and all questions were answered. The individuals who agreed to participate provided informed written consent. Overall, 29 participants were recruited. The participants were advised that their involvement was voluntary, that they could withdraw at any time, and that participation would not influence their care. A USD 25 gift card was provided as compensation for participation. The participants’ names and other demographic data were collected. They were also assigned a study number to protect confidentiality. The study number was securely kept with the audiotape and transcription data. The PI of the parent study maintained the list of participant names and demographics separately and securely.

The parent study collected the data using a Qualtrics online survey to obtain demographics and then used the Project Sickle Cell SBAR website for the intervention and post-intervention survey. The research team created this website for the parent study. The participants completed the study activities in a private area before or after their clinic appointment. The web-based training recorded the participants’ responses to prompts that simulated care seeking during an SCC in the emergency department or inpatient admission.

The study included web-based training, video vignettes, and coaching to teach the participants how to use SBAR as a communication tool in the emergency department or inpatient setting. The participants were given information and were educated about using each component of SBAR. They were shown a video of an interaction with and without using SBAR. After the video training, they were able to practice their skills. They were asked to imagine that they had presented to the ED during a crisis. The simulated HCP in the scripted video discussed an aspect of their care with the participant using each component of SBAR. The participants then had the opportunity to respond to the simulated HCP with the appropriate component of SBAR. They were given hints to help guide their answers. After the subjects completed the training, they were asked to provide feedback about the training using the prompt “Tell us anything you would like us to know about this communications skills website using video and the trainer.”

The current study involves the data collected during the emergency department simulation. Table 1 describes the emergency department HCP’s communication and a prompt to aid the participant in responding to the interaction using the appropriate SBAR component. Table 1 also outlines what information the participant was given to prompt their response to each component of SBAR.

The dataset consists of the transcribed recordings of their responses to each part of SBAR and the scenario provided. Five pieces of qualitative data for each participant were contained within the transcript. The five pieces included the recorded response for each component of SBAR (4) and the response to the post-intervention survey (1) that had the previously mentioned open-ended statement.

## 2. Materials and Methods

Qualitative and quantitative descriptive methods were used to complete the data analysis. Inter-rater reliability (IRR) of the qualitative data was used to evaluate the usefulness and accuracy of the participants in answering the prompts of the SBAR-cued communication responses. Content analysis was utilized to describe the participants’ perspectives on the acceptability of using the SBAR patient–HCP communication simulation.

Aim 1: Evaluate the usefulness and accuracy of the participants in answering the prompts of the SBAR-cued communication responses. To address aim 1, the current study used a coding system and IRR to determine whether the participants correctly used SBAR after the training. The IRR of the qualitative data, a theoretically supported coding system, was developed to guide the raters in analyzing each response [30]. Cohen’s kappa was used to determine the standard of agreement between the raters.

Aim 2: Describe the individuals’ perspectives on the acceptability of using the SBAR patient–HCP communication simulation to better prepare for ED visits during an SCC. To address aim 2, the PI of the current study used a qualitative descriptive approach, employing content analysis of the data from the open-ended post-training question, where the participants were asked to provide feedback. This analysis was guided through the lens of the study’s theoretical framework. The IRB review with a resulting exempt status was obtained through the University of Massachusetts Medical School (UMMS) before the secure data transfer. The transferred data included the transcribed audio recordings of the participants from the parent study.

### 2.1. Data Analysis 

Three raters used a theoretically based exemplar to determine whether the parent study participants answered the component of SBAR correctly by assigning a score to each response to conduct the IRR of the qualitative data. These best-practice exemplars were determined from the literature regarding the use of SBAR and the prompts provided to the participants [20,23]. Each rater read through each transcript and assigned a score, as explained below, based on whether the participant appropriately answered each component of SBAR. The raters used the participants’ answers to the SBAR prompts to determine whether the component of SBAR had been addressed as the participants had been coached to respond.

To determine the initial IRR, three participant transcripts were randomly selected and scored by the three raters to establish reliability and ensure consistent procedures between the raters, to answer questions, and to address any potential problems. Because this was the first time SBAR training was used for this population, it was essential to use the actual data from the participants for IRR as opposed to mock data. Mock data may not have been representative of the data in the pilot study. This initial scoring allowed each rater to compare their scoring using the best-practice exemplar to ensure that the instructions were understood. These data were not included in the final analysis.

To score the data, each component of SBAR was evaluated for each participant on a scale based on how the component of SBAR was answered. Zero (0) indicated that the participant did not respond using SBAR. A score of one (1) indicated they addressed the component of SBAR correctly. The scoring results were then analyzed using SPSS software to calculate the percent agreement between the raters. The researcher used pairwise deletion to manage missing data. Pairwise deletion allows the researcher to use all available data and omit any missing piece of data rather than deleting entire cases [31].

Qualitative content analysis was used to analyze the open-ended qualitative data. These open-ended data comprised the fifth piece of the data: the post-training question outlined in Table 2. This was an open-ended feedback question. The 1st and 4th authors completed the qualitative analysis and agreed upon the themes. They used the following process:The researcher read the entire transcript several times for data immersion;Codes were derived by identifying words that captured important concepts;Notetaking was conducted to capture and identify codes;Related codes were then categorized;These categories were further coded into meaningful groups;The categories were grouped into subcategories based on their relationships;These categories were defined;Data exemplars were identified to support the categories and subcategories.

### 2.2. Trustworthiness

This study used the standards of trustworthiness described by Lincoln and Guba [30]: credibility, transferability, dependability, and confirmability.

Credibility (which is similar to internal validity and the truth of data) ensures that the readers’ perspective aligns with the participants’ views, meaning that the interpretations are viewed similarly [30]. Due to this being a pilot study with limited access to participants, member checks were not conducted. However, credibility was enhanced with the usefulness of the web-based trainer based on the responses being evaluated using the SBAR component which was consistent with the qualitative data from participants. To discover coding biases, peer debriefing took place with co-authors 2 and 3, who were not involved in the parent study or qualitative analysis.

Transferability refers to whether the results of the analysis can be transferred to other settings and participants [30]. To achieve transferability, the researchers described the procedures and methods used in this study. Furthermore, a detailed description of the findings was presented. This was achieved by using direct quotations that supported the emerging themes.

Dependability refers to whether the findings are aligned with and supported by the data and whether they are consistent. Dependability ensures that the process used to analyze the data is sound and determines whether the process is clearly documented [32] so that others who might want to replicate the study can do so. The researcher ensured the planned study procedures were meticulously followed and kept a record of the procedure and data management by keeping an audit trail. These details ensure that the study can be replicated.

Confirmability refers to whether the results can be corroborated [32]. Using records and documentation during the study serves as the “audit trail” which will support confirmability.

## 3. Results

All the available data were included in the analysis. All the participants who met the inclusion criteria were included, and their demographic data can be found in Table 2. The participants’ ages ranged from 18 to 62 years old. Education level and gender demographics were also collected (Table 3). Of the 29 participants chosen for the parent study, 18 completed the modules; therefore, 18 available transcripts were used for IRR. Three participant transcripts were used for IRR and were not used in the final analysis. Ten participants responded to the open-ended question. There were missing data due to two participants completing the modules twice, participants leaving before they could finish the modules, or errors with recording the responses on the platform.

### 3.1. Inter-Rater Reliability

Out of the 18 available transcripts, 15 were used. The three transcripts used for the initial pilot study were not included in the final analysis. The IRR using percent agreement ranged between 64% and 95%. The IRR percent agreement among the participant responses regarding the “situation” component of SBAR was 89%. The IRR percent agreement among the participant responses regarding the “background” component of SBAR was 95%. The IRR percent agreement among the participant responses regarding the “assessment” component of SBAR was 82%. The IRR percent agreement among the participant responses regarding the “recommendation” component of SBAR was 64%. According to Cohen’s kappa, an acceptable level of IRR is 0.6 [33]. The percentage of participants who answered SBAR correctly based on the three raters’ scoring is outlined in Table 3.

The raters agreed to adhere to providing a “1” rating for any answer that partially answered the component of SBAR. Some participants responded to SBAR in the inappropriate section of SBAR, such as by providing R content during the S portion of the interaction. These were not considered as appropriately answering the expected component of SBAR as they may have indicated a lack of understanding of the component. Most of the participants, 94% and 95%, were able to describe the situation and background correctly, respectively. At 87%, fewer were competent with the assessment portion of SBAR. The lowest percentage correct was in the recommendation aspect of SBAR at 72%.

### 3.2. Qualitative Content Analysis

The participants were asked to provide feedback regarding what else they would like to share about the communication skills website using the videos and trainer. In addition, many of the participants also shared their ED experiences when interacting with the HCPs. The responses of ten participants were analyzed. 

Of the 10 participants who responded to the open-ended questions, four major categories were identified regarding the participants’ feedback when using the trainer. They included: (1) Patient–Provider Communication and Interaction; (2) Patients Want to be Heard and Believed; (3) the Accuracy of the ED Experience and Incorporating the Uniqueness of Each Patient; and (4) the Overall Usefulness of the Video Trainer. The data results supporting each category are presented below.

#### 3.2.1. Patient–Provider Communication and Interaction

Of the ten responses, four participants shared having similar experiences interacting with HCPs regarding their sickle cell disease progression and management, as presented in the vignette. They described HCPs as making dismissive assumptions about their reasons for seeking care. There were many comments about HCPs immediately assuming a person of color was a drug addict, which delayed care because the patient needed to convince the provider they required pain management. In addition, HCPs based some of their evaluations on observation and on the patients not looking like they were in great pain vs. considering variation in the physical presentation of pain. The participants described how HCPs (doctors and nurses) spoke to them as if the patients were lying about their pain. They shared that this HCP interpretation obstructs effective pain crisis treatment.


*“The way the nurse and doctor talked to her [in the trainer video] is exactly how they talked to you as they think that you are lying about how you are feeling.”*


The participants described being dismissed and not believed when seeking care in their own experiences. They even identified with some of the scenarios, such as the behavior not reflecting pain assessment and the commonality of the statements of the HCPs toward their disease process when they sought help.

#### 3.2.2. Patients Want to Be Heard and Believed

The second category that was identified by four of the ten participants was Patients Want to be Heard and Believed about their pain experience. Pain is the most common symptom in patients who have SCD, and many seek emergency care for pain crises. The participants wanted the providers to listen, understand, and believe their perspectives. 

The participants described feeling as if their words were less important because their labs and other diagnostics did not demonstrate the manifestation of disease as understood by the HCPs. The lack of the appearance of pain does not negate the experience of pain. These participants reported that because they do not appear to be in pain, it seems as if, from the HCP perspective, they do not have pain, which is incredibly frustrating. The participants explain that simply listening had the potential to improve the interaction and help the patient.

One participant stated:


*“Just because one does not look like he is in pain does not mean he does not feel it.”*


The participants describe the pain-relieving interventions they use to cope, such as music, talking on the phone, or trying to sleep. These adaptive coping measures are incongruent with the pain presentation expected by some healthcare providers. Because of this discrepancy, the coping mechanisms may be weaponized against them. One participant said that things started to “go wrong” when the nurse refused to listen to their concerns.

#### 3.2.3. Accuracy of the ED Experience and Incorporating The Uniqueness of Each Patient

The third category identified was Accuracy of the ED Experience Incorporating the Uniqueness of each Patient. Of the ten responses, four participants stated that they felt the trainer was accurate and helpful. In contrast, two felt the trainer did not accurately depict an ED care-seeking experience, as each patient is unique. The other two participants did not speak specifically about the accuracy of the trainer.


*“This thing hits all of us very different.”*


One stated that it was difficult to act as a patient as this was a simulated experience. The participant was not currently in pain or interacting with an actual provider, further illustrating that the trainer was not compelling for this participant. Although it is meant to demonstrate a real-world experience, the trainer is a simulation and cannot represent all patient experiences accurately.

Conversely, four participants stated that they felt the trainer accurately captured an SCC and the interaction with the HCP during a crisis. They went further and described this situation as similar to the situations that they and others had been in and that it represented a realistic view of the experience of a patient in SCC seeking care. One participant stated that not only did the vignette demonstrate the possibility of misunderstanding but also that the vignettes demonstrated how the patient could use SBAR to communicate their needs.

One of those four participants stated that although they could not relate to the patient because of the individual experience of a sickle cell crisis, they nevertheless agreed that the HCP depiction was accurate regarding how they spoke to the patient and dismissed their symptoms.

#### 3.2.4. Overall Usefulness of the Video Trainer

The final category is the Overall Usefulness of the Video Trainer. Six participants described the perceived usefulness of the trainer modules and communication tools. Of those who discussed the usefulness of the trainer, they overwhelmingly felt that the trainers were helpful and could facilitate communication. The participants thought SBAR could be used for more thorough communication with HCPs. The use of the trainer can help garner trust with the HCP. They describe how practicing with the recording helped them understand and apply the material to their own experiences. They felt the tool helped them to learn how to use SBAR.


*“I think SBAR is a great method for better and more thorough communication with health care providers.”*


## 4. Discussion

Patients with SCD face unique challenges during care seeking for pain management in emergency departments. These challenges have been described by patients and HCPs [4,6,7,8,10,13,14,15,16,18,19]. The current study results support using SBAR as a communication tool for patients to communicate with providers in the emergency department. The agreement among the raters illustrates the consistency across the three reviewers. This study demonstrates the need for better communication strategies for patients and providers and the ability to tailor this tool and teach patients how to communicate their needs.

The participants’ responses to the study illustrated the lack of understanding and empathy from HCPs. As pain is subjective and these patients are adaptable, typical observational manifestations of pain may be absent, leading to possible skepticism from providers regarding the patient’s pain. This possible skepticism makes patients feel like they are not being heard and that the providers simply are not listening. The participants’ descriptions are consistent with how vulnerability factors impact outcomes [7,10,11,26].

The findings of this study are supported by the theoretical framework (SCMSCD). SBAR can be considered as an assertive communication method—a self-care management resource that can positively influence health outcomes. The pilot demonstrated that individuals with SCD can learn SBAR and use it during interactions with HCPs. SBAR can help them collaboratively seek care while developing a helpful trusting relationship with HCPs. The mistrust in the SCD community, described by LaMotte and colleagues [15], may be enhanced by improved communication built on respect. Because HCPs already use SBAR as a communication tool, it may also enhance the credibility of patients with SCD [23]. This could lead to better and more informed care for these patients, which would lead to increased health outcomes, as theorized in SCMSCD. For example, if individuals with SCD are confident that they can communicate to obtain the care they need, they may respond differently to the cues related to their SCD (vulnerability factor). Additionally, assertive communication using SBAR will improve important outcomes such as health-related stigma. For example, by using SBAR, the patient with SCD can explain the coping mechanisms that an HCP may consider as drug-seeking behaviors. Communication is critical, given the results of one study [13] where HCPs reported that patient behavior was the highest barrier to care. More effective patient–provider communication may decrease this barrier and the health-related stigma endured by individuals with SCD.

Even with most participants agreeing that this tool is helpful, there is room for improvement. Some participants had difficulty responding due to their inability to relate their SCD crisis experiences to the trainer module. Providing more generalized prompts to a patient experience may be one way to mitigate this. Creating a scenario that captures every patient’s essence and unique attributes may be challenging. Thus, adapting the scenarios based on further exploration of the SCD pain crisis experience with the patients’ recommendations may bring more specificity to the learning experience.

This study has clinical implications and the ability to impact provider–patient communication. This study offers an initial framework that can help to create and lead to training programs to teach patients how to communicate with healthcare providers to facilitate improved outcomes effectively. Patients will have the skills to provide needed information regarding their health status and disease progression in a format healthcare providers understand and can use to treat them.

Although the tool was successful in helping the participants communicate their needs, some participants did not use it correctly. This may be attributed to the hints that were provided. Providing more general hints may help to tailor this activity to patients with SCD at every stage of care. It may also be important to conduct the study in a non-clinic environment where time can be allotted for completing it.

The IRR for the participant responses was high among the situation, background, and assessment components. However, in the recommendation component, the participants scored the lowest. Typically, patients are not given the opportunity to recommend or direct their care. This is typically the role of the provider who makes decisions regarding the patient’s care. Therefore, these patients were expected to have difficulty grasping this training area. However, patients are expected to contribute to recommendations in a more collaborative, patient-centered model. In the case of SCD, patients learn about their bodies, which can be beneficial to HCPs in arriving at a diagnosis and treatment plan. More attention should be given to the recommendation component to ensure that participants understand their role in their care.

Although this pilot study had many strengths, some limitations must be acknowledged. The sample size was small and was a convenience sample. Additionally, all the data were collected from one southeastern sickle cell program. For all of these reasons, these results are not generalizable. Additionally, the sample was predominantly female.

## 5. Conclusions

Communication between patients and healthcare providers is essential to increasing the outcomes, especially in marginalized communities. Using SBAR as a modality to teach patients to communicate may improve these outcomes. The challenge is creating a modality to teach patients these techniques in a meaningful, patient-centered way. Continued work and education regarding the use of SBAR with patients with SCD are needed, but this study provides a foundation for developing a teaching platform for patients. SBAR will need to be tailored to meet the needs of diverse individuals living with SCD. A web-based SBAR training model is an essential first step to using the gold standard of communication among HCPs in a nuanced way to benefit a vulnerable patient population. Improving patient–provider communication is an important step in mitigating racialized disparities in pain and enhancing trust for vulnerable patient populations, such as individuals with SCD [15,16].

## Figures and Tables

**Figure 1 ijerph-19-13817-f001:**
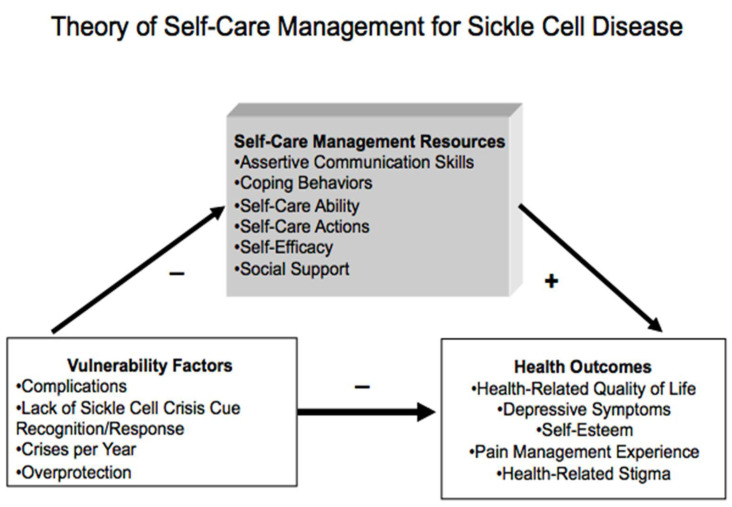
Theory of Self-Care Management for Sickle Cell Disease (SCMSCD).

**Table 1 ijerph-19-13817-t001:** SBAR Prompts and Hints.

SBAR Component	HCP Interaction/Statement	Hint for SBAR Response
Situation: Pretend you are a patient with SCD seeking care in the ED. You have a pain crisis. The doctor came in to evaluate you. This begins with some history of the present issue.	“What brought you in today?”	You will describe your reason in a few words. Include your SCD type and pain score on a scale from 0 to 10. Remember, you did not wait until your pain was a 9 or 10 before you sought care. Remember to include the location of your pain.
Background: The history continues and includes related history in addition to the current issue.	“Most sicklers come in with pain or 9 or 10. I need to examine you and decide what to do next.”	You will briefly describe the background. Remember to include anything about the typical course of your pain crises and what you tried before coming to the emergency department.
Assessment: The history continues, and the assessment begins.	“Your vitals look good. You say you already tried your pain pills at home?”	You will briefly describe what you think is going on. Include what you think is going on and anything about the typical course of your pain crisis.
Recommendation: The situation, background, and assessment are complete, and a determination of the next steps is needed.	“We try to see the sickest patient first, and you don’t look that bad. I noticed you were listening to music when I came in. You look pretty good.”	Include your recommendation for what you would like to happen next (e.g., pain medication or intravenous fluids). Briefly explain why you were listening to music (or whatever you might be doing for distraction). Add your goals for discharge and your desire to go home.

**Table 2 ijerph-19-13817-t002:** Demographic Data.

Education Level		Gender	
8th through 11th grade	3%	Female	68%
Completed High School or GED	29%	Male	32%
After High School Training	3%		
Some College	39%		
College Graduate	23%		
Postgraduate	3%		

**Table 3 ijerph-19-13817-t003:** Participants who answered SBAR Correctly.

Component of SBAR	Percentage of Participants Who Answered SBAR Correctly
Situation	94%
Background	95%
Assessment	87%
Recommendation	72%

## Data Availability

Data available on approved request.

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
