# Peer review of "Individuals with Sickle Cell Disease Using SBAR as a Communication Tool: A Pilot Study"

_ijerph, 2022, doi:10.3390/ijerph192113817_

Round 1
Reviewer 1 Report (Previous Reviewer 1)
This paper is novel and has great significance to the care of individuals with sickle cell disease. The revisions to the manuscript significantly improved the strength of the paper. Thank you for addressing much-needed items, particularly the informed consent and transparency of methodology. One last suggestion is to add how many patients were approached for the current study and how many declined, with the reason for the decline on p. 5, lines 189-196.
Author Response
Please see the attachment.

Reviewer 2 Report (Previous Reviewer 2)
While the authors have done a good job with the revision, I still see the need to review the very recent literature from 2022 and to include according studies into the intro and the discussion. Furthermore, the discussion needs to round up the whole study. The readability is still not high and the triangulation of the results is needed. Please consider a figure aggregating the findings and reintegrating them into the theoretical background you described in the into.
Furthermore, an extensive spell/grammar check is required and the formatting of the file to be improved.
Author Response
Please see the attachment.

This manuscript is a resubmission of an earlier submission. The following is a list of the peer review reports and author responses from that submission.
Round 1
Reviewer 1 Report
Interventions that focus on improving patient-provider communication for adults with sickle cell disease (SCD) are critical. The authors of the current study sought to develop and evaluate the feasibility of a SBAR intervention in 18 adults with SCD. The study has many strengths, namely the clever application of a well-known communication tool, mixed-method design, and use of the self-care management for SCD model. The manuscript could be strengthened further by developing a clearer justification for using the SBAR communication tool and more details in the methods section, specifically the development and implementation of the SBAR training to participants, the scoring procedure, and qualitative results. See my comments below for more information.
Introduction
• The Wakefield et al., 2018, referenced in the Bias and Stigma section, described community racial bias more than medical bias in youth with SCD qualitatively. A small subset of the sample discussed medical bias, but may not be enough evidence to be the only reference following the statement “racial bases play a significant role in this population’s care.” Additional racial bias literature would be helpful to add to this section.
• Because the SBAR communication tool is the focus of the intervention, additional discussion about the decision to use this strategy over other communication tools would strengthen section 1.3 or on page 4, second paragraph.
• Please explain the rationale for the waiver of written informed consent. Participants were asked to engage in study activities that may have warranted more informed consent procedures as it is described currently.
• More information would be helpful about the web-based training, video vignettes, and coaching procedures on page 5, lines 180-189. For example, how long did the training take? How much coaching was involved? Did the authors standardize the training or coaching session to account for potential attention biases?
Methods and Materials
• Please include a reference after the sentence on page 6 starting line 227 indicating the use of the ‘literature’ to determine SBAR best practices.
• Please describe the rationale for using three transcripts at random versus mock transcripts to help with rater training. This method reduced the sample size further, and information about any potential differences between the demographics of participants withdrawn is needed.
Results
• 18 of the 29 participants of the parent study completed the modules. Please share information on why 11 participants did not complete the modulus on p. 7 lines 291-293.
• Consider removing Table 4 because the findings are described in the text.
• Regarding the qualitative analyses, please clarify who conducted these analyses, if there was more than one coder, and if inter-rater reliability was determined.
• Information about how many participants described the Accuracy of the ED Experience and Incorporating The Uniqueness of each Patient theme, but it is unclear how many participants described the other three themes. Please include this information in these sections of the results (page 9-10)
Discussion
• The content of the Discussion section is appropriate and thoughtful. This section is missing a more detailed limitations section.
Reviewer 2 Report
This study investigates patients responses to SBAR categories, which is generally important.
While the study is purely qualitative, I see the merits of this approach. In general, more quantitative analyses could be added and a stronger connection of the findings with the theoretical background is needed:
The discussion needs significant improvement with regard to integrate into the theory is required. Please also consider giving the Figure 1 again in the discussion and integrating the findings into this figure.
Most importantly, the authors need to update their lit review with the years 2021 and 2022 as I could not find any of the relevant articles from the last 2 years.
Author Response
- While the study is purely qualitative, I see the merits of this approach. In general, more quantitative analyses could be added and a stronger connection of the findings with the theoretical background is needed:
- Further quantitative analysis would not address the study aims thereby deviating from the goals of the study
- The discussion needs significant improvement with regard to integrate into the theory is required. Please also consider giving the Figure 1 again in the discussion and integrating the findings into this figure.
- Added at line 426 - These self-care management resources also influence other factors in the continuum of care. This empowers patients to take a leading role in their care. Communication can have a positive influence on their self esteem and pain management experience as outlined in the SCMSCD.
- Most importantly, the authors need to update their lit review with the years 2021 and 2022 as I could not find any of the relevant articles from the last 2 years.
- THis author was unable to locate relevant articles dated in 2021 or 2022. However there are several articles dating from 2019 and 2020
